# The Integrated Disturbance Estimation and Non-Singular Terminal Sliding Mode Longitudinal Motion Controller for Low-Speed Autonomous Electric Vehicles

**DOI:** 10.3390/s25185799

**Published:** 2025-09-17

**Authors:** Boyuan Li, Wenfei Li, Wei Hua, Lei Guo, Haitao Xing, Hangbin Tang, Chao Huang

**Affiliations:** 1Hong Kong Center for Construction Robotics (InnoHK Center Supported by Hong Kong ITC), Hong Kong; boyuanli@hkcrc.hk; 2Research Centre for Intelligent Equipment, Zhejiang Laboratory, Hangzhou 311100, China; tanghangbin@zhejianglab.org; 3Research Centre for Intelligent Manufacturing Computing, Zhejiang Laboratory, Hangzhou 311100, China; guolei@zhejianglab.com (L.G.); h.xing@zhejianglab.com (H.X.); 4Faculty of Sicences, Engineering and Technology, University of Adelaide, Adelaide 5005, Australia; chao.huang@adelaide.edu.au

**Keywords:** disturbance estimation, non-singular terminal sliding mode control, model uncertainty, actuator delay

## Abstract

In the current literature, few motion control studies have considered the disturbances caused by road profile, model uncertainty, and actuator delay in regard to low-speed autonomous vehicles. In addition, motion controller designs usually rely on motor/brake torque control, which is not always available. This study outlines an integrated disturbance estimation and non-singular terminal sliding mode controller (NS-TSMC) to overcome disturbances in low-speed scenarios through traction/brake pedal position control. First, a longitudinal dynamic model that considers a detailed brake-by-wire hydraulic braking system model and a motor actuator model is proposed. Road disturbances, model uncertainty, and actuator delays are also considered in vehicle modelling. This vehicle model was verified through experimental data from a low-speed autonomous sightseeing vehicle. Then, based on the proposed vehicle model, the disturbance and uncertain parameter estimator was designed and integrated with NS-TSMC to achieve longitudinal motion control through throttle/brake pedal control. Experimental results from the experimental sightseeing vehicle and simulation results demonstrated the improvement of the longitudinal motion tracking performance and motion comfort compared with a benchmark proportional–integral–derivative (PID) longitudinal motion controller.

## 1. Introduction

Vehicle longitudinal motion control is crucial for highly automated modern vehicles. Ranging from anti-lock braking systems (ABSs), traction control, and simple cruise control to complex adaptive cruise control, different forms of vehicle longitudinal motion control have been widely applied in the highway scenario [1,2,3]. In highway conditions, disturbances caused by unknown parameters in traction or braking systems, vehicle state sensor measurement errors or road profiles can be neglected or easily compensated for using a classical linear feedback controller. However, these disturbances pose significant challenges during longitudinal motion in low-speed scenarios. Reliable longitudinal motion control for low-speed scenarios is critical for autonomous sightseeing vehicles, autonomous delivery vehicles, or autonomous automobile in urban driving conditions.

In the current literature, the longitudinal motion controllers examined usually have hierarchical structures, including high-level motion planners and low-level motion tracking controllers [4,5,6,7,8,9,10,11,12,13,14]. High-level longitudinal motion planners usually determine the desired longitudinal speed or acceleration profile through an optimisation algorithm based on the real-time information of other road participants and the environment, and some studies also considered the optimisation of energy efficiency [8] or passenger comfort [9]. The focus of our study is on low-level motion tracking controllers, so high-level motion planning controllers will not be discussed in detail.

For low-level longitudinal motion tracking controllers, an optimal speed-tracking controller based on model predictive control (MPC) is proposed to determine the motor control torque [8], and the proposed controller is validated using simulation results from a low-speed sightseeing vehicle. However, the low-level controller in [8] only determined the actuator control torque, and the switch control between the traction motor actuator and brake actuator was not discussed. In addition, the direct control of the motor or brake torque is usually unavailable for a motion tracking controller, since tier 1 chassis suppliers do not always open the channel, so the control input obtained from the throttle or brake pedal position is used instead. In [9], the low-level controller calculated the expected throttle opening and brake pressure of master cylinder according to the desired acceleration command. In order to avoid damage to the vehicle’s system, a switch strategy was designed to guarantee that the brake pedal and accelerator pedal do not function at the same time. The main issue of this study is that the uncertainty of the longitudinal traction and the braking model caused by road disturbances, model parameter uncertainty, or model system delay is not considered. Marcano et al. proposed a model of longitudinal vehicle dynamics using a detailed motor traction model and a braking system model in an experimental vehicle, i.e., a Renault Twizy, and the traction and braking actuator delays were also determined based on the road test results [10]. Based on the longitudinal dynamics model, three different control strategies, namely the proportional–integral–derivative (PID) controller, the adaptive network fuzzy interface system (ANFIS), method and the MPC method, are proposed to achieve the velocity tracking though the control inputs obtained from throttle and brake pedal positions [10]. The authors of [14] argue that road profile disturbances (such as steep slopes, speed bumps, and potholes) cause the classical PID feedback controller to become stuck or to over-shoot the set-point value for acceleration and thus greatly impair driving comfort and safety in low-speed and low-acceleration conditions; therefore, a low-level acceleration model predictive controller (MPC) was designed in [13] by utilising knowledge of road profile ahead to control the engine and brake proactively. Although MPCs have been widely applied in the motion control of autonomous vehicles, they are mainly used for multiple target optimisation problems with multiple constraints. However, the proposed longitudinal motion controller in this study only deals with a single control target—the desired longitudinal velocity—so the SMC is enough to solve this problem with a reduced computational cost compared with MPC.

As can be seen in this review of the current literature [15,16,17,18,19], unlike theoretical motion control studies which assume direct control of the motor or brake torque, the low-level longitudinal motion controller utilizes the detailed motor traction model and braking system model to determine the relationship between the throttle/braking pedal position and acceleration/braking torque. Secondly, the disturbance caused by the road profile, the parameter uncertainty model, and actuator delays should be considered in the dynamics modelling and addressed in the motion tracking controller design [20,21,22,23,24,25]. Thirdly, the low-level motion tracking controller design for low-speed scenario is less focused.

In order to solve the above issues, as outlined in the literature, the major contributions of our study can be summarised as follows: Firstly, we proposed a detailed longitudinal dynamics model by considering the detailed brake-by-wire hydraulic braking system model and the motor actuator model in low-speed scenarios. The disturbances caused by the road profile, actuator delays, and model parameter uncertainties were also considered. Secondly, a non-singular terminal sliding mode controller, taking into consideration the disturbance estimator, was proposed for low-level longitudinal motion tracking control in order to improve tracking performance through throttle/braking pedal position control.

The rest of this paper is organised as follows. In Section 2, the vehicle longitudinal dynamics model and experimental validation are presented. In Section 3, the sliding mode controller design is described. Section 4 provides the experimental and simulation results, and Section 5 gives the conclusions.

## 2. Longitudinal Vehicle Dynamics Model and Experimental Validation

In this section, the longitudinal vehicle dynamics model is presented, and the model parameters are calibrated and verified by the actual experimental results from a low-speed sightseeing autonomous vehicle. More importantly, the model utilizes the relationship between the acceleration or brake pedal position and the longitudinal tyre force, which provides the fundamental model for the motion controller design.

### 2.1. Longitudinal Vehicle Dynamics Modelling

The longitudinal vehicle dynamics model can be presented as the following equation [26]:(1)$${\dot{v}}_{x}=\frac{T_{t}-T_{b}-mgf_{res}R_{\omega }}{mR_{\omega }}-\frac{1}{2m}\rho_{0}AC_{d}v_{x}^{2}+g\sin\;\varphi$$
where $v_{x}$ is the vehicle’s longitudinal velocity (m/s), m is the vehicle mass (kg), $R_{\omega }$ is the wheel radius (m), $T_{t}$ is the traction torque generated by the traction motor (N), $T_{b}$ is the brake torque generated by hydraulic braking system (N), $f_{res}$ is the coefficient of the wheel rolling resistance, g is the gravity acceleration (m/s^2^), $\rho_{0}$ is the air density (kg/m^3^), A is the air drag front area (m^2^), $C_{d}$ is the wind drag coefficient, and φ is the road slope. It can be seen that aerodynamic drag has a very limited influence on the overall longitudinal dynamics in low-speed scenarios.

The traction motor of the experimental sightseeing vehicle is in the speed control mode, which means that the throttle pedal opening position corresponds to the specific desired longitudinal speed. When throttle opening is reduced, the traction motor will operate in the reverse mode to generate a brake torque effect in order to reduce the longitudinal speed. The traction torque model can be presented by the following equation [27,28]:(2)$$T_{t}=\frac{1}{\tau_{1}s+1}e^{-\tau_{2}s}\left(K_{p\_m}e_{v}+K_{I\_m}\int e_{v}\right)$$(3)$$e_{v}=v_{xd}\left(P_{cc}\right)-v_{x}$$
where $\tau_{1}$ represents the actuator delay of traction motor and $\tau_{2}$ is the pure delay caused by the signal transmission lag between the throttle pedal comment and the traction motor. $P_{cc}$ is the throttle pedal position, and the desired longitudinal velocity $v_{xd}\left(P_{cc}\right)$ can be determined as a function of $P_{cc}$. The PI controller in Equation (2) can be considered the speed controller of the traction motor actuator. The control error of the PI controller $e_{v}$ is the different between the target longitudinal speed and the actual vehicle speed. The selection of the $K_{p\_m}$ value can determine the convergence speed of the PI controller, and this convergence speed is determined by the motor angular acceleration. The actual traction motor angular acceleration has a specific maximum value due the motor power limit, so the $K_{p\_m}$ should not be set as a very large value.

The braking torque generated by the hydraulic braking system can be modelled using the following equation [29]:(4)$$T_{b}=\frac{4\mu_{b}A_{p}r_{0}p_{p}}{R_{\omega }m}\frac{1}{\tau_{3}s+1}e^{-\tau_{4}s}$$
where $\mu_{b}$ is the friction coefficient of the brake disc, $A_{p}$ is the cross area of the wheel cylinder, and $r_{0}$ is the effective radius of the friction brake, $p_{p}$ is the braking pressure of wheel cylinder, $\tau_{3}$ is the braking actuator delay caused by the hydraulic system, and $\tau_{4}$ is the pure delay cased by the space between braking rod and braking disc.

In order to determine the relationship between the brake pedal position $P_{pd}$ and the brake pressure of wheel cylinder, the brake pressure of the master cylinder and wheel cylinder can be modelled as shown below. The master cylinder pressure can be determined by the following equation:(5)$${\dot{p}}_{c}=\frac{\beta }{\left[L_{c}-x_{c}\left(P_{pd}\right)\right]A_{c}}\left(A_{c}{\dot{x}}_{c}\left(P_{pd}\right)-q\right)$$
where $p_{c}$ is the braking pressure of the master cylinder, $A_{c}$ is the cross area of the master cylinder, $L_{c}$ is the maximum stroke of the master cylinder, β is the bulk modulus of the brake fluid. $x_{c}$ is the displacement of the stroke of the master cylinder, which is the function of the brake pedal position $P_{pd}$. q is the volumetric flow rate running out of the master cylinder, which is equal to the volumetric flow rate running into the wheel cylinder. The wheel cylinder pressure can be determined as follows:(6)$${\dot{p}}_{p}=\frac{K_{p}q}{A_{p}^{2}}$$
where $K_{p}$ is the pad stiffness of the wheel cylinder. The volumetric flow rate can be calculated as follows:(7)$$q=\frac{p_{c}-p_{p}}{K_{pl}}$$
where $K_{pl}=32\upsilon \rho \pi \frac{L_{pl}}{A_{pl}^{2}}$, υ is the brake fluid viscosity, ρ is the fluid density, $L_{pl}$ is the length of the pipeline, and $A_{pl}=\frac{1}{4}\pi d_{pl}^{2}$ is the cross-section area of the pipeline.

### 2.2. Model Parameter Calibration

The experimental autonomous sightseeing vehicle for model parameter calibration is shown in Figure 1. The vehicle is a front-wheel steering and rear-wheel driving vehicle equipped with motor driving system and an electric hydraulic brake (EHB) system. In the model calibration process, some basic vehicle parameters are assumed to already be known and are shown in Table 1.

In the traction model in Equation (2), the pure delay caused by the signal transmission lag $\tau_{2}$ is determined by the lag between the acceleration pedal command and the measured longitudinal velocity, as shown in Figure 2.

The model calibration figure (as shown in Figure 3) is determined based on a set of longitudinal traction and brake manoeuvres in the field test. As can be seen in the traction mode of Figure 3, the desired longitudinal velocity $v_{xd}$ can be determined by a particular pedal position $p_{cc}$, which is shown in the lookup table in Table 2. It should be noted that the maximum longitudinal velocity is limited to 4.6 m/s in the experimental vehicle for calibration, as shown in the lookup table. $K_{p}$ and $K_{I}$ can be tuned and determined by comparing the model simulation results and the experiment measurement results of longitudinal speed in the pure traction motion, as shown in Figure 4a.

The displacement of the stroke of the master cylinder can be determined by the brake pedal position, $P_{pd}$, based on the model calibration; see Figure 3. As shown in the brake mode in Figure 3, the brake pedal position is related to the target acceleration regardless the vehicle longitudinal speed. Therefore, the calibration procedure for determining the above relationship can be designed as follows: first a lookup table is created based on the relationship between the brake pedal position and longitudinal acceleration; secondly, a lookup table describing the relationship between the longitudinal acceleration and the displacement of the stroke of the master cylinder is created based on Equations (1) and (4)–(7); finally, a lookup table of the relationship between the braked pedal position and the master cylinder stroke displacement can be created accordingly. This lookup table is shown in Table 3.

After all the other parameters in the brake mode have been calibrated and determined, the first-order delay of the hydraulic braking system $\tau_{3}$ can be determined by comparing the model simulation results and the experimental measurement results of the longitudinal speed, as shown in Figure 4b. The calibration parameters are listed in Table 4.

## 3. Sliding Mode Controller Design

Based on the proposed longitudinal dynamics model, this section focuses on the discussion and design of the longitudinal motion tracking controller by assuming that the desired longitudinal velocity is already known. Instead of the traction or brake torque control, which is not available in many commercial chassis systems, the controller inputs in this study are derived from the throttle or brake pedal positions. First, a simple benchmark PID longitudinal motion tracking controller is described for comparison purposes, and then the non-singular terminal SMC (N-S TSMC), including the unknown parameter and disturbance estimator, is proposed and discussed. The relevant control parameters are also listed in Table 4.

### 3.1. PI Controller Design

In the PI control design, it is assumed that the desired longitudinal velocity is known and planned using the path planner layer, and the actual longitudinal velocity can be assumed to be easily measured or estimated by the on-board sensors. The velocity tracking error ev can be described as follows:(8)ev=vxd−vx
where vxd is the desired longitudinal velocity. The designed PI controller has a traction control mode and a brake control mode. In the traction control mode, the throttle pedal position is considered the control input; while in the brake control mode, the brake pedal position is considered the control input. The switch logic for traction control and brake control is designed as Algorithm 1.
**Algorithm 1** The switch algorithm for traction and brake PI controllerIf ev_c<=eth or ev_c>0 (error ev_c is positive value or ev_c is smaller than the threshold value)ev_PI=maxKP_cev_c+KI_c∫ev_c,0 (Calculate the positive PI control values)vxd_lookup=vxd+ev_PI (Calculate the desired longitudinal speed)pcc=maxminPcc_lookupvxd_lookup,Pcc_max,0(Calculate the acceleration pedal position through lookup table)Else (If error ev_c is negative value and ev_c is larger than the threshold value)ev_PI=−(KP_cev_c+KI_c∫ev_c) (calculate the negative PI control values) Pbp=minPbp_lookupev_PI,Pbp_max(Calculate the brake pedal position through lookup table)
where eth is the constant value which determines the dead-zone range to disable the brake control when the error ev_c is not too big. KP_c and KI_c are PID control gains. Throttle pedal position Pcc_lookupvxd_lookup is the function of vxd_lookup, and this function is determined by the lookup table (Table 2). Pcc_max, Pbp_max is the maximum throttle and brake pedal position. The brake pedal position Pbp_lookupev_PI is the function of ev_PI and a lookup table can describe this relationship (Table 3).

**Remark** **1.**
*It should be noted that there are two PI controllers, one for the vehicle modelling and one for the vehicle longitudinal motion controller design, respectively. The PI controller in the vehicle model is used to approximate the behaviour dynamics of the driving motor, while the PI controller in the longitudinal motion control is utilised to track the desired vehicle longitudinal velocity.*


### 3.2. N-S TSMC Design

In the SMC design, the switching strategy of the acceleration and brake mode can be determined by the desired longitudinal acceleration $a_{xd}$. The Lypunov candidate function can be designed as follows:(9)$$\frac{1}{2}s^{2}=\frac{1}{2}{\left(\int e_{v}+\frac{1}{\beta_{1}}{\left(e_{v}\right)}^{\frac{p_{1}}{q_{1}}}\right)}^{2}$$
where $\beta_{1}$,$p_{1}$,$q_{1}$ are constant parameters for the SMC. $\beta_{1}>0$, $p_{1},q_{1}$ are positive odd numbers.

When the vehicle is in the acceleration mode ($a_{xd}>0$), the derivative of Equation (9) can be described based on Equations (1) and (2) by neglecting the traction actuator delay:(10)$$\begin{array}{ll} \frac{d\left(\frac{1}{2}s^{2}\right)}{dt} & =s\left(e_{v}+\frac{p_{1}{\dot{e}}_{v}}{q_{1}\beta_{1}}{\left(e_{v}\right)}^{\frac{p_{1}}{q_{1}}-1}\right)\\ & =s\left[e_{v}+\frac{p_{1}}{q_{1}\beta_{1}}{\left(e_{v}\right)}^{\frac{p_{1}}{q_{1}}-1}\left({\dot{v}}_{d}-\frac{T_{t}-\mu mgf_{res}R_{\omega }}{mR_{\omega }}-\frac{1}{2m}\rho_{0}AC_{d}v_{x}^{2}+g\sin\varphi \right)\right]\\ & =s\left[e_{v}+\frac{p_{1}}{q_{1}\beta_{1}}{\left(e_{v}\right)}^{\frac{p_{1}}{q_{1}}-1}\left({\dot{v}}_{d}-B_{1}\left(c_{1}P_{cc}+c_{2}\right)-{∆}_{1}-{∆}_{2}-{∆}_{3}\right)\right] \end{array}$$
where $B_{1}=\frac{1}{mR_{\omega }}$, $c_{1}$ and $c_{2}$ are the constant values representing the relationship between the throttle pedal position and traction torque. ${∆}_{1}$ and ${∆}_{2}$ are model disturbances caused by the rolling resistance, wind drag force, and road slope, where ${∆}_{1}=-\mu g\left(f_{res}+k_{res}v_{x}^{2}\right)$ and ${∆}_{2}=-\frac{1}{2m}\rho_{0}AC_{d}v_{x}^{2}+g\sin\varphi$. $1<\frac{p_{1}}{q_{1}}<2$. ${∆}_{3}$ is the disturbance term caused by the uncertain traction motor parameters.

The throttle pedal input can be designed as follows:(11)$$P_{cc}=\frac{K_{1}sgn\left(\frac{1}{\beta_{1}}{\left(e_{v}\right)}^{\frac{p_{1}}{q_{1}}}\right)-B_{1}c_{2}+{\dot{v}}_{d}+\frac{\beta_{1}q_{1}}{p_{1}}{\left(e_{v}\right)}^{2-\frac{p_{1}}{q_{1}}}}{B_{1}c_{1}}\;P_{cc}=max\left[min\left(P_{cc},1\right),0\right]$$

Equation (10) can be rewritten as following equation according to the control law (11):(12)$$\frac{d\left(\frac{1}{2}s^{2}\right)}{dt}=\frac{p_{1}}{q_{1}\beta_{1}}{\left(e_{v}\right)}^{\frac{p_{1}}{q_{1}}-1}\left\{-K_{1}\left[sgn\left(\frac{1}{\beta_{1}}{\left(e_{v}\right)}^{\frac{p_{1}}{q_{1}}}\right)\int e_{v}+\left|\frac{1}{\beta_{1}}{\left(e_{v}\right)}^{\frac{p_{1}}{q_{1}}}\right|\right]-\left({∆}_{1}+{∆}_{2}+{∆}_{3}\right)s\right\}$$

Since $e_{v}>0$ in the traction mode, $\frac{1}{\beta_{1}}{\left(e_{v}\right)}^{\frac{p_{1}}{q_{1}}}>0$, $sgn\left(\frac{1}{\beta_{1}}{\left(e_{v}\right)}^{\frac{p_{1}}{q_{1}}}\right)>0,\int e_{v}>0$. When $K_{1}$ is selected as a positive value so that $K_{1}\left[sgn\left(\frac{1}{\beta_{1}}{\left(e_{v}\right)}^{\frac{p_{1}}{q_{1}}}\right)\int e_{v}+\left|\frac{1}{\beta_{1}}{\left(e_{v}\right)}^{\frac{p_{1}}{q_{1}}}\right|\right]+\left({∆}_{1}+{∆}_{2}+{∆}_{3}\right)s>0$, $\frac{d\left(\frac{1}{2}s^{2}\right)}{dt}$ is always negative. It should also be noted that the throttle position is constrained within [0, 1]. If the value of $K_{1}$ is too big, the pedal control input will go over the boundary and the control performance will be affected.

In order to avoid the chatting effect of the SMC, sgn(x) is replaced by the saturation function sat(x):(13)$$sat(s)=\left\{\begin{array}{cc} sgn(s) & if\;\left|s\right|>∆\\ \frac{2s}{∆} & else \end{array}\right.$$
where ∆ is the constant parameter which determines the boundary of transition layer.

When the vehicle is in the brake mode ($a_{xd}\leq 0$), the derivative of Equation (10) can be described as neglecting the brake actuator delay:(14)$$\frac{d\left(\frac{1}{2}s^{2}\right)}{dt}=s\left[e_{v}+\frac{p_{1}}{q_{1}\beta_{1}}{\left(e_{v}\right)}^{\frac{p_{1}}{q_{1}}-1}\left({\dot{v}}_{d}-\frac{-T_{b}-\mu mg{f_{res}R}_{\omega }}{mR_{\omega }}-\frac{1}{2m}\rho_{0}AC_{d}v_{x}^{2}+g\sin\varphi \right)\right]\;=s\left[e_{v}+\frac{p_{1}}{q_{1}\beta_{1}}{\left(e_{v}\right)}^{\frac{p_{1}}{q_{1}}-1}\left({\dot{v}}_{d}-B_{2}\left(c_{3}P_{bp}L_{c}+c_{4}\right)-{∆}_{1}-{∆}_{2}-{∆}_{4}\right)\right]$$
where $B_{2}=-\frac{4\mu_{b}A_{p}r_{0}}{R_{\omega }m}$, and $c_{3}$ and $c_{4}$ are the constant values representing the relationship between the brake pedal position and brake pressure of wheel cylinder. ${∆}_{4}$ is the disturbance term caused by the uncertain brake motor parameters.

The brake pedal input can be designed as follows:(15)$$P_{bp}=\frac{K_{2}sgn(\frac{1}{\beta_{1}}{\left(e_{v}\right)}^{\frac{p_{1}}{q_{1}}})-B_{2}c_{4}+{\dot{v}}_{d}+\frac{\beta_{1}q_{1}}{p_{1}}{\left(e_{v}\right)}^{2-\frac{p_{1}}{q_{1}}}}{B_{2}c_{3}L_{c}}\;P_{bp}=max\left[min\left(P_{cc},1\right),0\right]$$

Equation (14) can be rewritten as the following equation, according to the control law (15):(16)$$\frac{d\left(\frac{1}{2}s^{2}\right)}{dt}=\frac{p_{1}}{q_{1}\beta_{1}}{\left(e_{v}\right)}^{\frac{p_{1}}{q_{1}}-1}\left\{-K_{2}\left[sgn\left(\frac{1}{\beta_{1}}{\left(e_{v}\right)}^{\frac{p_{1}}{q_{1}}}\right)\int e_{v}+\left|\frac{1}{\beta_{1}}{\left(e_{v}\right)}^{\frac{p_{1}}{q_{1}}}\right|\right]-\left({∆}_{1}+{∆}_{2}+{∆}_{4}\right)s\right\}$$

Since $e_{v}<0$ in the brake mode, $\frac{p_{1}}{q_{1}\beta_{1}}{\left(e_{v}\right)}^{\frac{p_{1}}{q_{1}}-1}>0$, $sgn\left(\frac{1}{\beta_{1}}{\left(e_{v}\right)}^{\frac{p_{1}}{q_{1}}}\right)<0$, $\int e_{v}<0$. When $K_{2}$ is selected as a positive value so that $K_{2}\left[sgn\left(\frac{1}{\beta_{1}}{\left(e_{v}\right)}^{\frac{p_{1}}{q_{1}}}\right)\int e_{v}+\left|\frac{1}{\beta_{1}}{\left(e_{v}\right)}^{\frac{p_{1}}{q_{1}}}\right|\right]+\left({∆}_{1}+{∆}_{2}+{∆}_{4}\right)s>0$, and $\frac{d\left(\frac{1}{2}s^{2}\right)}{dt}$ is always negative. Similarly, if $K_{2}$ is too big, the control performance will be compromised.

### 3.3. N-S TSMC Design with Uncertain Parameter and Disturbance Estimator

In the longitudinal motion, the disturbance parameters are hard to accurately estimate, such as the rolling resistance, wind drag coefficient, or road slope. In addition, certain traction motor and brake actuator parameters and actuator delays are unknown or hard to estimate, and the traction or brake actuators may become worn. Thus, including uncertainty parameters and disturbance estimators in the SMC design can greatly improve control performance.

The estimator can be designed as the following equation [10]:(17)$$\dot{\hat{x}}=Au+\hat{\sigma }+\frac{\alpha_{1}}{\varepsilon }(x-\hat{x})\;\dot{\hat{\sigma }}=\frac{\alpha_{2}}{\varepsilon^{2}}(x-\hat{x})$$
where x is the measured state and $\hat{x}$ is the estimated state of the estimator. $\alpha_{1}$,$\alpha_{2}$ are positive values and satisfy Hurwitz:(18)$$s^{2}+\alpha_{1}s+\alpha_{2}=0$$

ε can be determined as follows:(19)$$\frac{1}{\varepsilon }=\left\{\begin{array}{cc} 100t^{3} & 0\leq t\leq 1\\ 100 & t>1 \end{array}\right.$$

When the vehicle is in the traction mode, (17) can be rewritten as follows:(20)$${\dot{\hat{v}}}_{x}=B_{1}\left(c_{1}P_{cc}+c_{2}\right)+\hat{\sigma }+\frac{\alpha_{1}}{\varepsilon }(v_{x}-{\hat{v}}_{x})\;\dot{\hat{\sigma }}=\frac{\alpha_{2}}{\varepsilon^{2}}(v_{x}-{\hat{v}}_{x})$$
where the estimated disturbance value is $\hat{\sigma }={\hat{∆}}_{1}+{\hat{∆}}_{2}+{\hat{∆}}_{3}$. Similarly, when the vehicle is in brake mode, (17) can be rewritten as follows:(21)$${\dot{\hat{v}}}_{x}=B_{2}\left(c_{3}L_{c}P_{bp}+c_{4}\right)+\hat{\sigma }+\frac{\alpha_{1}}{\varepsilon }(v_{x}-{\hat{v}}_{x})\;\dot{\hat{\sigma }}=\frac{\alpha_{2}}{\varepsilon^{2}}(v_{x}-{\hat{v}}_{x})$$
where the estimated disturbance value is $\hat{\sigma }={\hat{∆}}_{1}+{\hat{∆}}_{2}+{\hat{∆}}_{4}$.

Based on (20) and (21), the throttle and brake pedal control law (11) and (15) can be updated as follows:(22)$$P_{cc}=\frac{K_{1}sgn\left(\frac{1}{\beta_{1}}{\left(e_{v}\right)}^{\frac{p_{1}}{q_{1}}}\right)-B_{1}c_{2}+{\dot{v}}_{d}-w_{1}\hat{\sigma }+\frac{\beta_{1}q_{1}}{p_{1}}{\left(e_{v}\right)}^{2-\frac{p_{1}}{q_{1}}}}{B_{1}c_{1}}\;P_{bp}=\frac{K_{2}sgn(\frac{1}{\beta_{1}}{\left(e_{v}\right)}^{\frac{p_{1}}{q_{1}}})-B_{2}c_{4}+{\dot{v}}_{d}-w_{2}\hat{\sigma }+\frac{\beta_{1}q_{1}}{p_{1}}{\left(e_{v}\right)}^{2-\frac{p_{1}}{q_{1}}}}{B_{2}c_{3}L_{c}}$$
where $w_{1}$,$w_{2}$ are the reduced factors of estimated disturbance value working together with $K_{1}$, $K_{2}$ to prevent the control input from passing the constraints. The stability proof is presented in Appendix A.

## 4. Experimental and Simulation Results

In this section, the experimental results of the PID longitudinal controller based on the experimental autonomous vehicle in Figure 1 is presented first, and the simulation results of the PID controller is compared with the experimental results in order to validate the simulation model and simulation performance. Then the simulation results of proposed N-S TSMC are compared with the PID to demonstrate the advantages of proposed approach.

### 4.1. Experimental Results

The experimental autonomous vehicle was equipped with an INS/GPS fusion system to measure the vehicle’s position and longitudinal velocity. A motion planning and PID longitudinal motion control strategy was implemented, and the desired velocity and traction (or brake pedal) input signals were recorded. The simulation model is a calibrated vehicle longitudinal dynamics model based on the experimental vehicle in Section 2. The tyre–road friction coefficient of the simulation model is assumed to be 1 as the experimental vehicle is driving on an asphalt road in good condition.

Figure 5a compares the velocity control performance by the PID controller between experimental results and simulation results, and the actual longitudinal velocity in simulation is similar to the results of the experiment. Figure 5b,c also shows that the traction pedal and brake pedal inputs in the simulation are similar to the inputs in actual experimental vehicle. All these results validate the vehicle model, and the PID controller in the simulation validates the actual PID control performance in the experimental vehicle. Thus, in the following section, the simulation results of PID controller can be used as benchmarks to validate the advantages of using the proposed N-S TSMC in a simulation environment.

### 4.2. Simulation Results of Proposed N-S TSMC

In this section, three sets of simulations are presented to compare the control performance between the proposed N-S TSMC and the PID by using the simulation software of Matlab Simulink 2015b (MathWorks, 1 Apple Hill Drive, Natick, MA 01760-2098, USA). In the first set of simulations, the simulation results of the same scenario as the experimental results are presented. In the second set of simulations, the $K_{p\_m}$ is reduced (from 70 to 30) to simulate the reduction in the maximum traction motor angular acceleration. In the third set of simulations, the road slope is set as 1.72 degrees (0.03 rad) to simulate an uneven road disturbance.

Figure 6 presents the longitudinal control performance in the first set of simulations. The velocity tracking performance of the PID controller is slightly better than the SMC alone and the SMC with an estimator (as shown in Table 5). At the beginning of the simulation, the longitudinal acceleration of the SMC alone and the SMC with an estimator increases to 1 m/s^2^ then drops gradually, which is more reasonable than the acceleration response of PID (increases to 0.5 m/s^2^, then immediately drops to 0, then increases to 0.8 m/s^2^, and after that gradually drops). Figure 7a,b shows the acceleration and brake pedal inputs for the speed tracking control. The pedal inputs of the PID show the same trend as the pedal inputs of SMC alone and the SMC with estimator. The pedal inputs of the SMC with an estimator have less oscillations than the pedal inputs of the SMC. This is because the proposed estimator can accurately estimate the disturbance value $\hat{∆}$ (as shown in Figure 7c; delta_acc means $\hat{∆}$ in acceleration mode and delta_brake means $\hat{∆}$ in braking mode) and less control effort is required to achieve a similar control performance (as shown in Table 6 and Table 7). Table 4 suggests that the control gain of the SMC is bigger than that of the SMC with an estimator. According to Figure 6 and Figure 7, PID shows comparable or even slightly better longitudinal control performance compared with the SMC alone and the SMC with an estimator, because the PID control gains are tuned based on this set of simulations.

Figure 8 shows the longitudinal control performance in the second set of simulations. Since the maximum motor acceleration is reduced, it takes a longer time for the driving motor to achieve the desired longitudinal velocity, as shown at the beginning of the simulation. Compared with the SMC alone and the SMC with estimator, the PID control shows worse speed tracking control performance during a period of 40–50 s when the vehicle is braking (as shown in Table 5). Figure 9 also suggests large oscillations of the pedal control inputs. These results can be explained as follows: the control gains of PID control are tuned based on the first set of simulations, and the control performance is compromised when the traction motor parameters change. On the other hand, the SMC is more robust to the uncertainty of the motor parameters and shows better control performance than the PID control, and the SMC with an estimator can further reduce the pedal control oscillations with less control effort (as shown in Table 6 and Table 7).

Figure 10 shows the longitudinal control performance in the third set of simulations, where the road slope disturbance and vehicle mass variance (the vehicle mass changes from 1490 kg into 1290 kg at 20 s) are considered. Although the PID control shows better velocity tracking performance in first 10 s compared with the SMC alone and the SMC with an estimator (as shown in Table 5), the acceleration response of the PID control has large oscillations. Figure 11 also demonstrates that the pedal control inputs have large oscillations. The PID control performance is greatly affected by the road slope disturbance, while the SMC alone and the SMC with an estimator still maintain a good control performance.

In order to further verify the proposed SMC performance in extreme conditions, the fourth simulation test was carried out during the emergency braking scenario. The total noise with a variance of 0.1 m/s was added into the measured longitudinal velocity to verify the robustness of the proposed controller. In Figure 12a, the proposed SMC and the SMC with an estimator show better velocity tracking control performances compared with PID control (as shown in Table 5), while Figure 12b suggests that the longitudinal acceleration is close to −6 m/s^2^, and the acceleration responses of the SMC alone and the SMC with an estimator exhibit fewer oscillations compared with PID control.

The fifth scenario is still the emergency braking but on an icy road with a friction coefficient of 0.3. It should be noted that since a low-speed scenario is assumed in this study, we did not consider longitudinal and lateral tyre friction forces. Therefore, the friction coefficient does not affect the acceleration capability in the vehicle model (1), which is fine for moderate traction and braking in low-speed scenarios. However, in emergency braking conditions with high accelerations, this assumption is not corrected and we must set a limit of the maximum acceleration based on the friction coefficient value. In Figure 13a, the proposed SMC and the SMC with an estimator show better velocity tracking control performances compared with the PID control (as shown in Table 5), but the velocity tracking performances are compromised when compared with Figure 12a due to the limitations of maximum braking acceleration. Figure 13b suggests that braking acceleration cannot achieve −3 m/s^2^, and the acceleration responses of the SMC alone and the SMC with an estimator display less oscillation compared with that of the PID control.

## 5. Conclusions

In this study, an integrated disturbance estimator and an N-S TSMC longitudinal motion controller are proposed to overcome parameter uncertainty and disturbance for low-speed automated vehicles. Based on the simulation results, the main findings can be summarized as follows:(1)The actual experiment was carried out to validate the proposed longitudinal vehicle dynamics model and the proposed PID controller, and the PID control can be used as the benchmark method for comparison with the control performance of the SMC.(2)When the PID control is well tuned in the first set of simulations, the PID control shows similar or even better longitudinal velocity tracking performance compared with the SMC.(3)When the simulation scenario changes, the longitudinal motion SMC is more robust with regard to parameter uncertainty and disturbances compared to the PID control and can achieve a better performance in terms of speed tracking, longitudinal acceleration, and smooth pedal control inputs.(4)For the proposed integrated disturbance estimator and the N-S TSMC approach, the disturbance estimator can successfully estimate the uncertainty parameters and disturbances, and the control effort can be reduced while maintaining good control performance.(5)In the emergency braking and icy road scenarios, the proposed SMC has better control performance in terms of longitudinal velocity tracking and acceleration.

In the future, we will improve our proposed N-S TSMC by considering the energy and comfort aspects and solving the issues of control input saturation and potential discontinuities. The proposed controller will be tested in a 3D simulation environment and using an experimental automated sightseeing vehicle to further validate control performance.

## Figures and Tables

**Figure 1 sensors-25-05799-f001:**
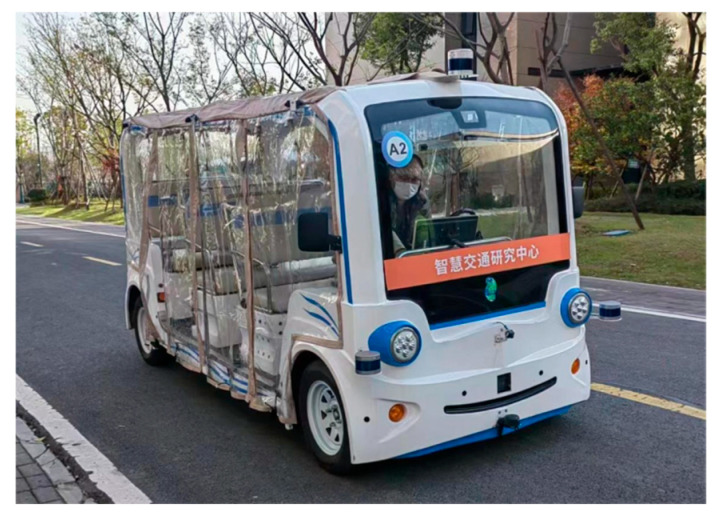
The experimental sightseeing vehicle. (“智慧交通研究中心” means Research center for intelligent transportation).

**Figure 2 sensors-25-05799-f002:**
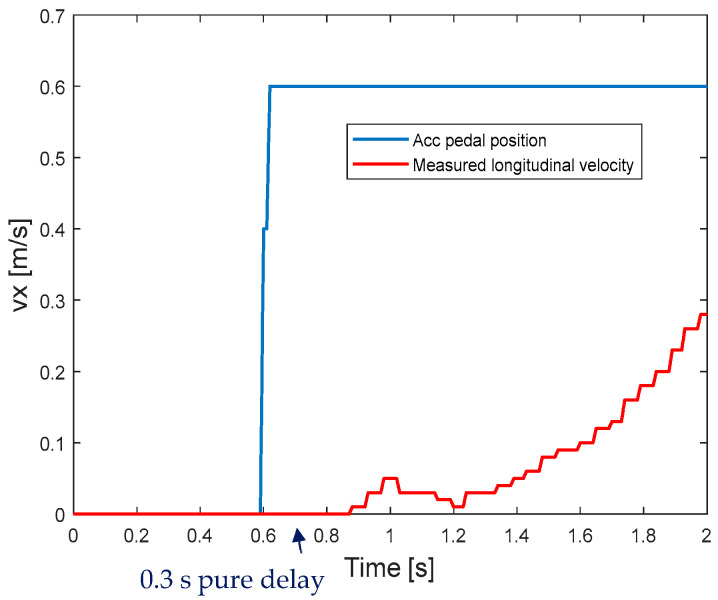
The measured longitudinal velocity and acceleration pedal position (0–1) in the field test.

**Figure 3 sensors-25-05799-f003:**
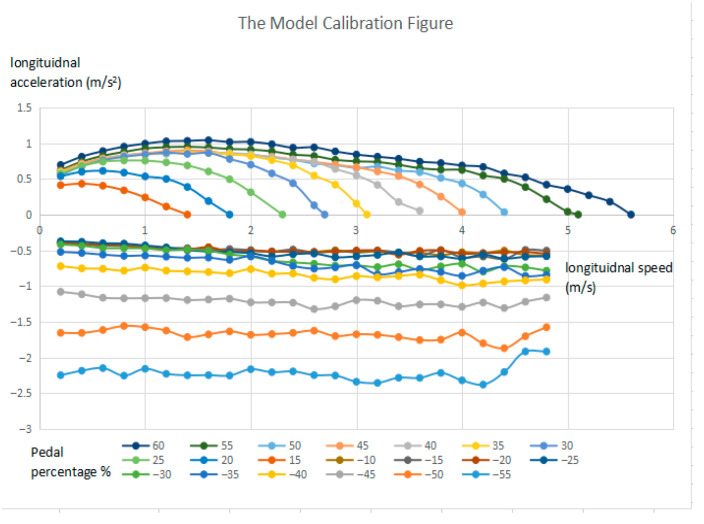
The model calibration figure based on the field test results.

**Figure 4 sensors-25-05799-f004:**
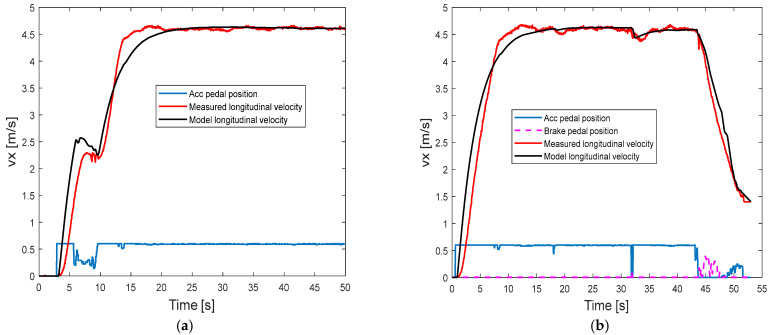
Compare the model longitudinal velocity and measured longitudinal velocity: (**a**) in pure traction mode; (**b**) in combined traction and brake mode.

**Figure 5 sensors-25-05799-f005:**
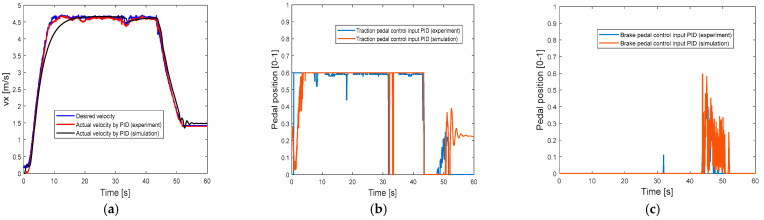
Compare the performances of PID controller in the experiment and the simulation: (**a**) velocity control performance of PID controller; (**b**) traction pedal input of PID controller; (**c**) brake pedal input of PID controller.

**Figure 6 sensors-25-05799-f006:**
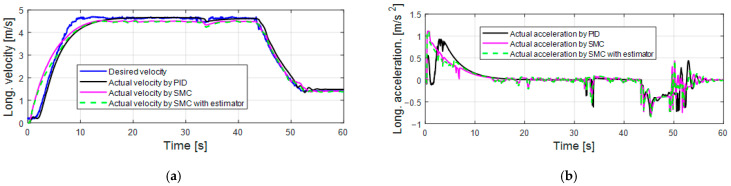
Longitudinal control performance in the first set of simulations: (**a**) longitudinal velocity; (**b**) longitudinal acceleration.

**Figure 7 sensors-25-05799-f007:**
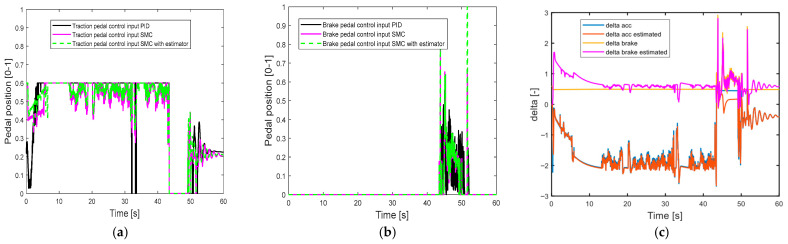
Pedal position in the first set of simulations: (**a**) traction pedal position; (**b**) brake pedal position; (**c**) uncertain disturbance estimation performance.

**Figure 8 sensors-25-05799-f008:**
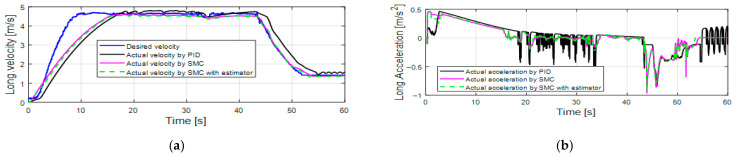
Longitudinal control performance in the second set of simulations: (**a**) longitudinal velocity; (**b**) longitudinal acceleration.

**Figure 9 sensors-25-05799-f009:**
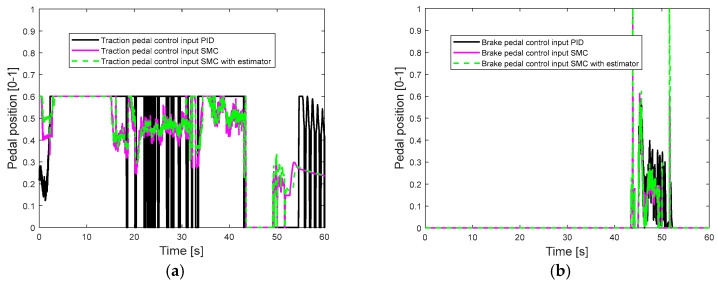
Pedal position in the second set of simulations: (**a**) traction pedal position; (**b**) brake pedal position.

**Figure 10 sensors-25-05799-f010:**
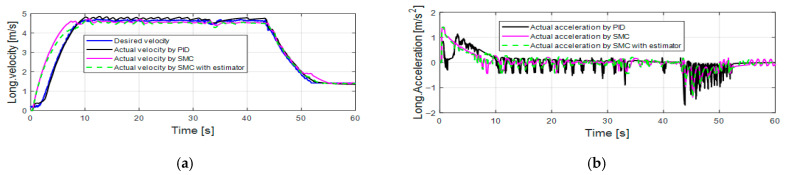
Longitudinal control performance in the third set of simulations: (**a**) longitudinal velocity; (**b**) longitudinal acceleration.

**Figure 11 sensors-25-05799-f011:**
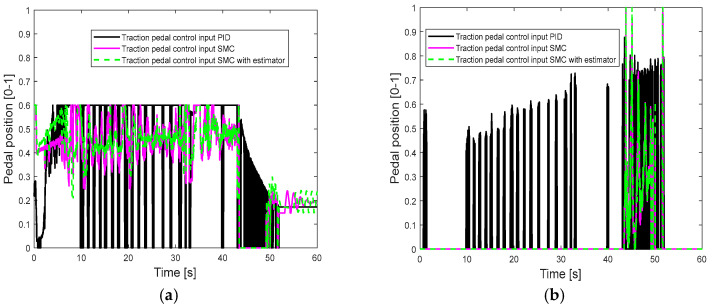
Pedal position in the third set of simulations: (**a**) traction pedal position; (**b**) brake pedal position.

**Figure 12 sensors-25-05799-f012:**
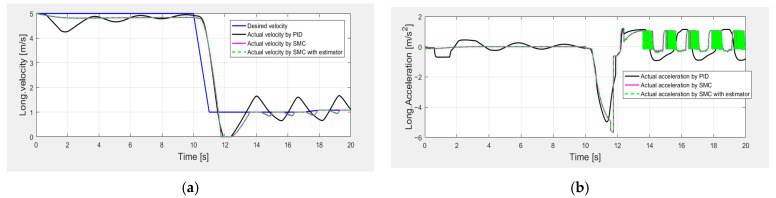
Longitudinal control performance in the fourth set of simulations: (**a**) longitudinal velocity; (**b**) longitudinal acceleration.

**Figure 13 sensors-25-05799-f013:**
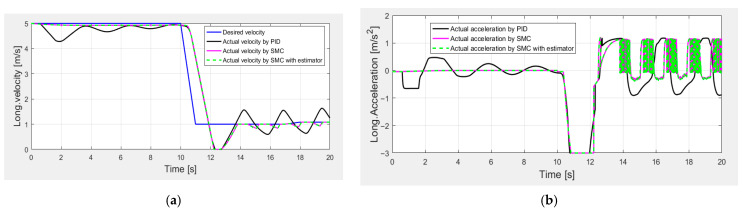
Longitudinal control performance in the fifth set of simulation: (**a**) longitudinal velocity; (**b**) longitudinal acceleration.

**Table 1 sensors-25-05799-t001:** The basic vehicle parameters of experimental sightseeing vehicle.

Symbol	Meaning	Values	Unit
m	Vehicle total mass	1490 (180 kg for 3 passengers)	kg
$R_{\omega }$	Tyre radius	0.165	m
μ	Tyre-road friction coefficient	1	N/A
$f_{res}$	Coefficient of wheel rolling resistance	0.011	N/A
$k_{res}$	Coefficient of wheel rolling resistance	$6.5\times 10^{-7}$	N/A
$\rho_{0}$	Air density	1.225	kg/m^3^
*A*	Air drag, front area	2.5	m^2^
$C_{d}$	Air drag coefficient	0.24	N/A
$\mu_{b}$	Friction coefficient of braking disc	1	N/A
$r_{0}$	Effective radius of friction brake	0.15	m
$L_{c}$	Maximum stroke of the master cylinder	25	mm
$A_{c}$	Cross area of the master cylinder	$3.5\times 10^{-4}$	m^2^
β	Bulk modulus of the brake fluid	$3.76\times 10^{6}$	N/A
$A_{p}$	Cross area of the wheel cylinder	$3.0\times 10^{-4}$	m^2^
$K_{p}$	Pad stiffness of the wheel cylinder	$1.0\times 10^{6}$	N/m
υ	Brake fluid viscosity	0.0025	m^2^/s
ρ	Fluid density	1000	kg/m^3^
$L_{pl}$	Length of the pipeline	1	m
$d_{pl}$	Diameter of the pipeline	0.008	m
$\tau_{1}$	Delay of the traction motor	0.025	s
$\tau_{4}$	Pure delay caused by the space between braking rod and braking disc	0.05	s

**Table 2 sensors-25-05799-t002:** Lookup table for the relationship between the acceleration pedal position and desired velocity.

Acceleration Pedal Position (%)	Desired Velocity (m/s)
0	0
10	0.6
15	1.2
20	1.7
25	2.2
30	2.6
35	3.1
40	3.6
45	4
50	4.3
55	4.4
60	4.6

**Table 3 sensors-25-05799-t003:** The lookup table for the relationship between brake pedal position, desired acceleration, and desired stroke displacement of the master cylinder.

Brake Pedal Position (%)	Desired Acceleration (m/s^2^)	Desired Stroke Displacement of Master Cylinder (mm)
0	0	0
−10	−0.416	1.200
−15	−0.418	1.232
−20	−0.420	1.248
−25	−0.432	1.264
−30	−0.473	1.280
−35	−0.573	1.760
−40	−0.742	2.320
−45	−1.169	3.680
−50	−1.575	4.800
−55	−2.158	6.400
−75	−4.230	12.800
−100	−5.000	16.000

**Table 4 sensors-25-05799-t004:** The calibrated vehicle parameters and controller parameters of the experimental sightseeing vehicle.

Symbol	Meaning	Values	Unit
$\tau_{2}$	Pure delay caused by the signal transmission lag	0.3	s
$\tau_{3}$	First-order delay of the hydraulic brake system	0.4	s
$K_{p\_m}$	P control gain of PI controller of traction motor	70	N/A
$K_{I\_m}$	I control gain of PI controller of traction motor	2	N/A
$K_{P\_c}$	P control gain for PID vehicle velocity controller	10	N/A
$K_{I\_c}$	I control gain for PID vehicle velocity controller	0.5	N/A
$K_{1}$	SMC gain of vehicle velocity controller in acceleration mode	15 (with estimator) 25 (without estimator)	N/A
$K_{2}$	SMC gain of vehicle velocity controller in brake mode	30	N/A

**Table 5 sensors-25-05799-t005:** Root mean square error (RMSE) of the velocity tracking error of different velocity controllers.

Simulation Test	PID Controller (m/s)	VeloSMC (m/s)	SMC with Estimator (m/s)
1	0.1690	0.2406	0.2284
2	0.5538	0.3786	0.4112
3	0.1038	0.3572	0.3092
4	0.5997	0.5470	0.5466
5	0.6287	0.5790	0.5788

**Table 6 sensors-25-05799-t006:** The average traction control pedal position [0–1] for different velocity controllers.

Simulation Test	PID Controller	SMC	SMC with Estimator
1	0.45	0.43	0.42
2	0.47	0.41	0.40
3	0.42	0.37	0.36
4	0.75	0.73	0.72
5	0.75	0.73	0.72

**Table 7 sensors-25-05799-t007:** The average braking control pedal position [0–1] for different velocity controllers.

Simulation Test	PID Controller	SMC	SMC with Estimator
1	0.07	0.07	0.07
2	0.07	0.07	0.07
3	0.20	0.11	0.11
4	0.40	0.39	0.38
5	0.42	0.40	0.40

## Data Availability

Data are contained within the article.

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
