# Peer review of "The Integrated Disturbance Estimation and Non-Singular Terminal Sliding Mode Longitudinal Motion Controller for Low-Speed Autonomous Electric Vehicles"

_sensors, 2025, doi:10.3390/s25185799_

Round 1
Reviewer 1 Report
Comments and Suggestions for Authors
Dear authors, please check the text of the article to identify irrelevant insertions.

Reviewer 2 Report
Comments and Suggestions for Authors
In this article, the authors propose an integrated disturbance estimation and non-singular terminal sliding mode controller (NS-TSMC) for the longitudinal control of low-speed autonomous electric vehicles. However, I will comment on some aspects to improve this manuscript.
- The authors only present simulated results, but detailed experimental evidence of the NS-TSMC is lacking.
- The authors should include all acronyms used with this "Anti-lock Braking System (ABS)" format.
- The authors do not comment on practical limitations such as actuator wear or extreme road conditions.
- Tables and figures should be as close as possible to where they have been referenced.
- The authors' proposal assumes constant friction coefficients, which is not reflected in real-life conditions where variability exists, such as wet or uneven pavement.
- The authors do not analyze the impact on controller stability when calibrating actuator delay.
- The authors do not make comparisons with other advanced methods or techniques such as MPC or LQR, which are commonly used in vehicle control.
- The authors do not evaluate the computational consumption of NS-TSMC for a real-time implementation.
- In line 249, it should be presented as an algorithm; please review how to do this (https://www.overleaf.com/learn/latex/Algorithms).
- The authors do not quantify the estimation error in real-life scenarios.
- The authors have not performed tests on abrupt disturbances such as emergency braking, as they have only been simulated on moderate slopes and with motor gain reduction.
- The authors do not provide metrics to compare controller performance.
- All figures should have the same format; some appear to have been developed with a specific format, while others appear to be screenshots.
- The authors should improve their conclusions and future work.
In this article, the authors propose an integrated disturbance estimation and non-singular terminal sliding mode controller (NS-TSMC) for the longitudinal control of low-speed autonomous electric vehicles. However, I will comment on some aspects to improve this manuscript.
- The authors only present simulated results, but detailed experimental evidence of the NS-TSMC is lacking.
- The authors should include all acronyms used with this "Anti-lock Braking System (ABS)" format.
- The authors do not comment on practical limitations such as actuator wear or extreme road conditions.
- Tables and figures should be as close as possible to where they have been referenced.
- The authors' proposal assumes constant friction coefficients, which is not reflected in real-life conditions where variability exists, such as wet or uneven pavement.
- The authors do not analyze the impact on controller stability when calibrating actuator delay.
- The authors do not make comparisons with other advanced methods or techniques such as MPC or LQR, which are commonly used in vehicle control.
- The authors do not evaluate the computational consumption of NS-TSMC for a real-time implementation.
- In line 249, it should be presented as an algorithm; please review how to do this (https://www.overleaf.com/learn/latex/Algorithms).
- The authors do not quantify the estimation error in real-life scenarios.
- The authors have not performed tests on abrupt disturbances such as emergency braking, as they have only been simulated on moderate slopes and with motor gain reduction.
- The authors do not provide metrics to compare controller performance.
- All figures should have the same format; some appear to have been developed with a specific format, while others appear to be screenshots.
- The authors should improve their conclusions and future work.
Reviewer 3 Report
Comments and Suggestions for Authors
This manuscript presents a novel strategy for longitudinal motion control with both theoretical development and real-world validation. However, some problems still need to be addressed as follows.
- The disturbance estimation results should be plotted since the disturbance estimator is listed as one of main contributions.
- The comparison of control efforts should be displayed quantitatively since the conclusion says the proposed control algorithm could save control effort.
- The stability proof should be provided for the disturbance observer based sliding mode control.
- All contents in one table should be displayed in one page.
- How does the authors address the inevitable uneven road profile and normally the vehicle mass variation, such as presented in doi: 10.1109/TITS.2025.3565857. in the proposed control strategy?
- How does the authors address the drawbacks of sliding mode control, such as the chatting issue and input constraints in the proposed control strategy?
Reviewer 4 Report
Comments and Suggestions for Authors
- The work is not limited to a simple TSMC N-S controller, but integrates an estimator for disturbances (road profile, actuator delays, parametric uncertainties). This combination increases robustness and reduces oscillations in the pedal commands (see Fig. 7 and Fig. 9).
- Unlike many theoretical works that assume direct control of engine/brake torque, here we work with the accelerator/brake pedal position. This is a pragmatic aspect, but it introduces a problem of nonlinearity and additional delays, which justifies detailed modeling.
- A complete model for the pressure in the master cylinder and the wheel cylinder is provided (equations 4a–4c), including the stiffness of the brake pad, the viscosity of the fluid, and the dimensions of the pipe. Such details are often missing in the literature.
- The model includes both pure delays (τ2, τ4) and first-order delays (τ1, τ3), implemented directly in the torque and braking dynamics equations. This is important for the stability of the controller and reflects experimental data.
- The following are tested: (i) nominal parameters, (ii) reduction of the maximum motor acceleration (from Kp_m = 70 to 30), and (iii) road slope of 1.72°. These demonstrate how the PID loses performance in the face of uncertainties, while the N-S TSMC remains stable.
- In Section 3.2 and the Appendix, Lyapunov functions are used to demonstrate the convergence of the error, with conditions on the gains K1/K2 for a negative sign in the derivative of the function. In the estimator, the Hurwitz condition is used for the choice of α1, α2.
- The authors emphasize that there are two PIs: one in the model (for the motor) and one in the controller (for tracking). Many readers might confuse these two roles without this explicit clarification.
- Simulations show that the PID was optimized only for the nominal scenario, losing performance in scenarios with parametric variations. The lack of a systematic method for adaptive retuning to the PID is a limitation.
- Although the PID is validated experimentally, the N-S TSMC is only simulated. The authors acknowledge this in the conclusions, but a validation on the vehicle would be crucial to demonstrate the benefits in practice.
- Although the introduction mentions that some works target energy efficiency and passenger comfort, the proposed assessment does not include energy or comfort metrics (jerk, RMS acceleration).
- Although equation (1) includes aerodynamic drag (Cd, A, ρ0), at low speeds its influence is minimal. It is not explicitly discussed whether the relevance of this term for the scenario up to 4.6 m/s has been verified.
- The current strategy involves switching between acceleration and braking modes, with dead zones. A blended control could eliminate potential discontinuities and improve the response to small disturbances.
- Equations (8) and (12) limit the commands Pcc, Pbp to [0, 1], but the stability proofs in Section 3 and Appendix do not explicitly include the effect of this saturation. In practice, saturation can lead to the loss of the finite-time convergence property of the N-S TSMC.
- The formulation (16) of ε(t) is a time-discontinuous function in the derivative at t = 1, which can introduce a small numerical impulse in discrete implementations. The analysis in Appendix mentions the peak error for small ε, but does not quantify the trade-offs with respect to sensor noise.
- The model includes pure and first-order delays (τ1–τ4), but the N-S TSMC design assumes “neglecting” them in the Lyapunov derivations (see 3.2, “neglecting actuator delay”). If the delays increase beyond the calibrated values (0.3–0.4 s), instability or oscillations may occur.
- The entire demonstration assumes that the longitudinal velocity vx is known exactly (via INS/GPS). In reality, GPS has noise of ±0.05–0.1 m/s, and filtering it out can introduce additional delays not included in the model, affecting the accuracy of the estimator and control.
Round 2
Reviewer 4 Report
Comments and Suggestions for Authors
The article has been adjusted according to the observations, and most of the suggestions have been addressed appropriately by the authors.
It is also necessary to consider the aspect related to the tables, many of which need adjustment because they either do not have enough information or simply require expansion, as they give the feeling that something is missing.
